# Climate Change and Human Response to Sustainable Environmental Governance Policy: Tax or Emissions Trading?

Qinglong Wang [1] , Jiale Huang [2], Xian Zhang [3,*], Weina Qin [4,*], Huina Zhang [5] and Yani Dong [6]

1 School of Marxism, Chengdu University of Technology, Chengdu 610057, China; wangqinglong@cdut.edu.cn
2 School of Business, Chengdu University of Technology, Chengdu 610057, China; huangjiale1@stu.cdut.edu.cn
3 School of Marxism, Sichuan University, Chengdu 610065, China
4 School of Marxism, Southwest Petroleum University, Chengdu 610500, China
5 Department of Cyber Economics and Management, Tellhow Animation College, Nanchang 330200, China; hn_zhang0505@yeah.net
6 China Telecom Corporation Lanzhou Branch, Lanzhou 730000, China; dongyn.gslz@chinatelecom.cn
* Correspondence: zhangxian@scu.edu.cn (X.Z.); 2017326250010@stu.scu.edu.cn (W.Q.)

**Abstract:** With climate change, humans are looking for effective ways to improve the ecological environment and provide comfortable survival space for sustainable development. There are two main economic methods for controlling environmental pollution: emissions fees (Pigouvian taxes) and emissions standards (emissions trading). However, the two policies have their applicable conditions in dealing with different sources of pollution and dynamic ecosystems, and many problems will arise if they are misused. This paper theoretically proves that significant (minor) pollution sources could satisfy the condition that the benefit function's curvature is greater (less) than that of the cost function. When the speed of ecological absorption is constant, a price policy controlling significant pollution sources will generate uncertainty; the quantity policy will generate a higher total social cost of managing the minor pollution source. When the speed of ecological absorption is not constant, adjusting the number of part pollution permits will lead to two kinds of pollutant leakage. If the pollution permits can be freely circulated, it will lead to the pollution of regional *b* (less regulated areas) inflow into the region *a* (more regulated areas); if permits are not freely circulated, otherwise not.

**Keywords:** climate change; human response; environmental governance policy; Pigou tax; emissions trading

## 1. Introduction

Since Marshall (1890) proposed the notion of internal economy and external economy, many economists have turned their attention to a market failure caused by externalities [1]. Before Coase's theory, the Pigou tax has been the primary economic means to regulate negative externalities. After Coase put forward their theory, the Pigou tax was challenged by emissions trading based on Coase's theory, and two kinds of governance means were formed in practice.

In 1968, Dales combined the emission standards and Coase theory (means) and first proposed the emissions trading theory in his book: Pollution, wealth, and price [2]. In addition, the US environmental protection agency (EPA) and Europe adopted their emissions trading theory to control pollution. Then, Carbon trading is employed in a wide range of countries, including Japan, South Korea, Kazakhstan, Switzerland, and China. Although various countries widely use emissions trading to control pollution, there is no disagreement about whether it affects pollution control. Richard (2017) analyzed more than three decades' experience in implementing emissions trading and believed that the design and implementation of total control and emission trading system are proper [3]. It can achieve its core goal of pollution control. However, the problem lies in details. First of all, the design of the system implementation and the economic environment are of

great importance. The setting of emission caps, testing and punishment for violations, and avoiding price fluctuations are crucial to emission trading. Secondly, the initial allocation of emission standards and emission leakage affects the government's selection of tools. These problems result in defects in the emission trading system. At the same time, some scholars believe that there are also some problems with the Pigou tax. On the one hand, the Pigou tax avoids the fluctuation of emission prices in emission trading; the optimal tax on emission collection brings problems for the implementers of policies. On the other hand, the Pigou tax does not recap total emissions, which is a disadvantage.

As previous studies had focused on price vs. quantity, which was more efficient in controlling environmental pollution, recent researchers studied applicable conditions of price and quantity under mixed pollution and stock pollution. Therefore, based on the above studies, this paper put forward three questions: (1) Is Pigou tax or emissions trading still effective in controlling environmental pollution from a dynamic perspective? (2) Which are more advantageous to control the significant or minor sources of pollution? (3) When the speed of ecological absorption is not constant. Do emissions trading still have relative advantages? This paper tries to study Pigou tax vs. Emissions trading from a dynamic perspective and propose an application range to control significant and minor pollution sources.

The rest of the paper is as follows. Section 2 reviews the literature. Section 3, referring to the model of Alpha C. Chiang, proposes the optimal amount of emission from a dynamic perspective. Section 4 studies applicable conditions of price or quantity to control a significant (minor) source of pollution. Finally, Section 5 provides a summary.

## 2. Literature Review

In the push to tackle climate change, price and quantity are the two main economic tools for human beings to deal with environmental pollution. Still, the dispute between them, which is more effective in controlling environmental pollution, has never stopped. The focus of its controversy mainly focuses on the following three aspects.

The first aspect compares a Pigovian tax and Coase theorem, which more efficiently regulates negative externalities. Pigou (1920) thought that it generated external uneconomic when the private marginal cost was smaller than the social marginal cost and proposed to solve it through taxation (Pigou tax) [4]. In Coase's (1960) paper, the problem of social cost, he criticized the communal principle of Pigou from the perspective of cause and effect, in which taxation by those responsible for externalities would distort the optimal allocation of resources [5]. He believed that as long as property rights were clear. The externalities represented by the damage would be self-corrected through the market; meanwhile, Coase Theorem laid the foundation for the subsequent emissions trading. During the dispute, most economists pointed out many defects in Pigou's theory, such as Pigou's tax's applicability under monopoly. Until Baumol and Oates (1971), they used the mathematical model contrast Pigou method and Coase method, which found that under the condition of Pareto optimality, causing external enterprise product price was equal to the marginal social cost ($p_1 = c_{11} + c_{21}$, $c_{11}$ was the marginal private cost of the enterprise, $c_{12}$ was the marginal external cost) [6]. After internalization, the product price was identical to the marginal personal cost. To get the actual price under the condition of perfect competition, it would need to be a tax on externalities products, namely, $t = c_{21}$. This view denied Coase's market-driven approach and supported Pigou's tax theory. In 1972, the environment committee of the Organization for Economic Co-operation and Development (OECD) first proposed the "polluter Pays Principle" (PPP principle), and the emissions fee was recognized and applied in various countries. Therefore, the Pigouvian taxes were commonly referred to as emissions fees.

The second aspect is mainly about the effective comparison between Pigou tax (price) and emission trading (quantity) [7–15]. When Dales proposed emissions trading, economists focused on whether the government-controlled the price (the Pigovian tax) and let the market determine the number of emissions. The government controlled

the quantity and let the market determine the price (emissions trading). Weitzman (1974) pointed out that the comparative advantage of setting quantity or setting price depended on the uncertain nature of the marginal benefits and costs of reducing pollution [16]. Still, it had considerable political appeal for the regulator (environmental protection department) to choose the quantity of emission pollution, indirectly supporting Dales' opinion. Later research in this aspect mainly expands the hypothesis of the model. For example, Laffont (1977) added the technology and information gap research based on the Weitzman model [17]. Browning (1987) thought that central institutions should consider regulation before considering the use of price or quantity to complete the plan and expanded the application scope of the Weitzman model by comparing price, quantity, and laissez-faire ranking [15]. Finkelshtain and Kislev (1997) compared the impact of price and quantity on governance externalities from a political perspective [11]. Ambeca and Coria (2013) studied the conditions of price and quantity selection under multiple pollutions [18]. Meunier (2018) analyzed whether unregulated externalities affect the choice between price and quantity tools [8].

The third aspect is mainly carbon tax and carbon trading, which are more suitable for reducing global carbon emissions [19,20]. The study of stock pollution, which was signed by the United Nations framework convention on climate change (FCCC) in 1992 and the Kyoto protocol in 1997, focused on things from emission charges (or Pigou taxes) and emission rights trading to carbon taxes, energy taxes, and carbon trading [21,22]. For example, Schelling (1997) believed that the emission reduction of carbon emissions might be an international political thorny issue, and the allocation of fair emission quotas, initially determined through negotiation, was questioned on its feasibility [23]. Akerlof (2006) supported the method of the carbon tax to reduce emissions. He believed that the carbon tax would impact people's behavior and could avoid the competition for emission rights in carbon trading [24]. Nordhaus (2007) proposed replacing the emission permit system with a carbon tax [25]. Parry and Pizer (2007) compared and contrasted policy approaches of carbon taxes and carbon trading policy approaches, such as price volatility, certainly emissions level, least-cost emissions reductions, raising revenue, incentivizing for R&D in clean technologies, practical or political obstacles, and new institutional requirements [26]. Dominioni (2022) argue that motivated reasoning contributes to this opposition by inducing the public to underestimate the effectiveness of carbon pricing to mitigate climate change and yield co-benefits [27].

There is a relatively large literature about the Pigou tax or emissions trading. Still, their studies have the following shortcomings: First, this literature mainly analyzes the problem that arises after implementing the two policies but does not answer the reasons for such issues. However, the two policies have their applicable conditions in dealing with different sources of pollution, and many above problems will arise if they are misused. Secondly, most of them only consider the role of static ecosystems and ignore the joint part of dynamic ecosystems, so the evaluation of policies is not accurate enough.

Based on the deficiencies of existing literature, the novelty of our approach might be manifest in several aspects:

- Firstly, we use the mathematical model to prove the applicable conditions of price policy instruments (Pigouvian tax) and quantity policy instruments (emission trading) to control large and small pollution sources.
- Secondly, we analyze the problems caused by two policies in the dynamic ecosystem. The price policy instrument (Pigouvian tax) will produce enormous social costs. The quantity policy instrument (emission trading) will have pollution leakage.
- Thirdly, the paper proposes to perfect the Pigouvian tax with a two-part charging to make up for the defects of the two policies under the dynamic mechanism.

## 3. Model

The optimal control theory model below mainly refers to Chiang (2016) [28]. In the process of fuel consumption, energy will be consumed, and a large amount of pollution

will be generated. $E$ is using an amount of energy. According to the nature of accumulation, we divide pollution into flow and stock pollution. $S$ is the stock pollution, and $S'$ is the flow pollution. Assume that there are $h$ polluters, and the pollution is directly proportional to energy use, setting $\varepsilon$, $0 \leq \varepsilon \leq 1$, is the scaling factor, which is expressed as:

$$S' = \varepsilon E \tag{1}$$

At the same time, the pollution $S$ is absorbed at speed $\delta$, $0 \leq \delta \leq 1$.

$$-S' = -\delta \varepsilon E \tag{2}$$

The changes in flow pollution are below the formula:

$$S' = \varepsilon E - \delta \varepsilon E \tag{3}$$

The use of energy $E$ increases the utility level of society, but at the same time, pollution $S$ is generated time 0 to $T$, and the accumulated pollution $S_T$ increases the negative utility of society. The utility function of society depends on the consumption and accumulation of pollution. The consumption function and the accumulation function of pollution are, respectively:

$$C = C(E) \quad C' > 0, C'' < 0 \tag{4}$$

$$S' = \varepsilon E - \delta \varepsilon E, S' > 0, S'' > 0 \tag{5}$$

The utility function of the society is

$$U = U[C(E), S(E)] \tag{6}$$

In addition, it has the following reciprocal Energy consumption can bring positive societal utility, and the more consumption brings, the greater the utility. In contrast, reducing energy storage can bring adverse societal utility, and the less the storage brings, the smaller the utility:

$$U_C > 0, U_S < 0, U_{CC} > 0, U_{SS} < 0, U_{CS} = 0 \tag{7}$$

The problem for the department of environmental protection is to maximize social utility in a given period $[0, T]$:

$$Max \int_0^T U[C(E), S(E)]dt$$

$$s.t. S' = \varepsilon E - \delta \varepsilon E$$

$S(0) = S_0, S(T) = S_T$, $S_0$ is the stock pollution at time 0, $S_T$ is the stock pollution of time $t$, and $S_0$, $S_T$ are constants. Constructing Hamilton function:

$$H = U[C(E), S(E)] + \lambda[\varepsilon E - \delta \varepsilon E] \tag{8}$$

Maximize the first-order condition:

$$\frac{\partial H}{\partial E} = U_C C'(E) + U_S S'(E) + \lambda(\varepsilon - \varepsilon \delta) \tag{9}$$

$$\lambda' = -\frac{\partial H}{\partial S} = 0 \text{ get } \lambda(t) = c \text{ (constant)} \tag{10}$$

The cross-sectional condition is

$$\lambda(T) \geq 0, 0 < S(T) < S_{ds=0}, \lambda(t)S(T) = 0, S(T) > 0$$

Suppose $\lambda(t) = 0$, (As the public does not pay for the air they use or the manufacturers do not pay for the emissions they emit into the air, the shadow price of the air they use is assumed to be zero.) the function of energy consumption satisfies the following conditions, $U_C C'(E) + U_S S'(E) = 0$. In the formula, the first item $U_C C'(E)$ measures the marginal positive utility taken from consumption by energy use $E$. The second item $U_S S'(E)$ measures the marginal negative effect taken from the accumulation of pollution from time $0$ to $T$. As the above equation $E$ is independent of the time variable $t$, the solution of optimal energy use is constant:

$$E^*(t) = E^* (E^* \text{is constant}) \tag{11}$$

The environmental protection department should balance the marginal positive and negative utility by choosing $E^*(t) = E^*$. Because the use of energy is constant, the integral over the equation of motion by, $S' = \varepsilon E - \delta \varepsilon E$

$$S^*(t) = (\varepsilon E^* - \delta \varepsilon E^*)t + k \tag{12}$$

Substitute $S(0) = S_0$ into (12), the optimal state path is

$$S^*(t) = (\varepsilon E^* - \delta \varepsilon E^*)t + S_0 \tag{13}$$

Any moment $S^*(t)$ depends on the speed of energy use $E$, the factor of pollution emission $\varepsilon$, the rate of ecological absorption $\delta$, time $t$, and $S_0$. The pollution, at time $T$ is

$$S_T = (\varepsilon E - \delta \varepsilon E)T + S_0 \tag{14}$$

According to Equation (14), there are three conditions for the pollution at the time $T$:

- If the speed of pollution discharge is greater than the speed of ecological absorption $\varepsilon E > -\delta \varepsilon E$, the pollution will accumulate based on $S_0$, $S_T > S_0$.

- If the speed of pollution discharge is less than the speed of ecological absorption $\varepsilon E < -\delta \varepsilon E$, the initial cumulative pollution $S_0$ will be absorbed by ecological $S_T < S_0$.

Therefore, when $S_T > S_0$, environmental protection departments need to control flow pollution, and the optimal control of the pollution makes the speed of pollution discharge equal to the rate of ecological absorption speed, that is

$$-S' = S' \tag{15}$$

Controlling the speed of pollution discharge depends on the rate of ecological absorption.

## 4. Price vs. Quantity on Controlling Significant (Minor) Pollution Sources

In the model below, we use the cost function $C(q, \theta)$ ($\theta$ is the disturbance term) and the benefit function $B(q, \eta)$ ($\eta$ is the disturbance term) of Weitzman's model. In addition, it satisfies $B''(q) < 0$, $C''(q) > 0$, $B'(0) > C'(0)$, if $q$ large enough, $B'(q) < C'(q)$, $q$ is clean air, $-q$ is the air pollution (The word "commodity" is used in an abstract sense and really could pertain to just about any kind of good from pure water to military aircraft. Solely for the sake of preserving a unified notation, we follow the standard convention that goods are desirable. This means that rather than talking about air pollution, for example, we instead deal with its negative—clean air [16]. Suppose $q_b$ is the pollution amount discharged by significant pollution sources, $q_s$ is the pollution amount discharged by minor pollution sources, and $q_b \gg q_s$, but there are the same the revenue function.

$$B(q, \eta) = B(\hat{q}, \eta) + [B' + \beta(\eta)](q - \hat{q}) + \frac{B''(q - \hat{q})^2}{2} \tag{16}$$

$C_b(q,\theta)$ is the cost function of a significant pollution source, $C_s(q,\theta)$ is the cost function of a minor pollution source, $C_b(q,\theta) > C_s(q,\theta)$.

$$C_b(q,\theta) = C_b(\hat{q},\theta) + (C' + \alpha(\theta))(q - \hat{q}) + \frac{C''(q - \hat{q})^2}{2} \tag{17}$$

$$C_s(q,\theta) = C_s(\hat{q},\theta) + (C' + \alpha(\theta))(q - \hat{q}) + \frac{C''(q - \hat{q})^2}{2} \tag{18}$$

Because $B''(q) > 0$, $C''(q) < 0$, when, $q_1 < q_2$,

$$B''(q_1,\eta) < B''(q_2,\eta), C''(q_1,\eta) < C''(q_2,\eta), (q_1 < q_2) \tag{19}$$

In conclusion, when Equation (19) establish, the curvature of the revenue function of significant pollution sources is greater than that of its cost function. The curvature of the cost function of minor pollution sources is greater than that of the benefit function. Therefore, the quantity policy tool is the first choice to control significant pollution sources, while the price policy tool is the first choice to control minor pollution sources (See Figure 1).

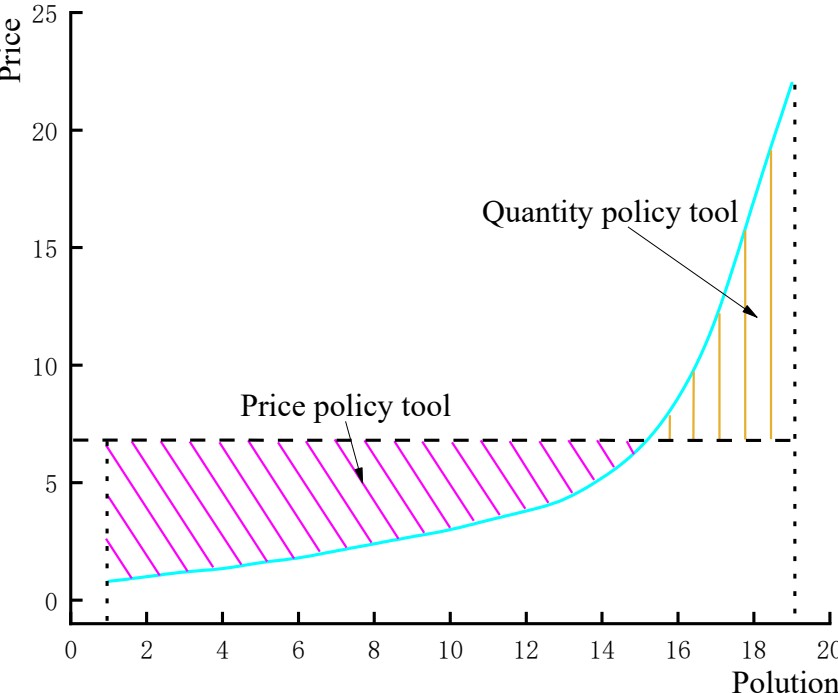

**Figure 1.** Price vs. Quantity.

Proposition: When the ecological absorption speed is constant, the quantity policy (emission standard) is the preferred tool to control significant pollution sources. The price policy (emission charge) is the selected tool for minor pollution sources.

### 4.1. The Uncertainty of Price (Pigou Tax) on Control Pollution

Emissions fees cannot control pollution to reach the optimal discharge level under the asymmetric information because the environmental protection department doesn't accurately estimate pollution's cost and benefit curve. Therefore, it is difficult for the environmental protection department to achieve the optimal emissions fee, making the emission reduction reach the ideal level $S_T^*$. In the model below, we use the cost function $C(q,\theta)$ ($\theta$ is the disturbance term) and the benefit function $B(q,\eta)$ ($\eta$ is the disturbance

term) of Weitzman's model. In addition, it satisfies $B''(q) < 0, C''(q) > 0B'(0) > C'(0)$, if $q$ large enough, $B'(q) < C'(q))$

$$C(q,\theta) = C(\hat{q},\theta) + (C' + \alpha(\theta))(q - \hat{q}) + \frac{C''(q-\hat{q})^2}{2} \tag{20}$$

$$B(q,\eta) = B(\hat{q},\eta) + [B' + \beta(\eta)](q - \hat{q}) + \frac{B''(q-\hat{q})^2}{2} \tag{21}$$

The corresponding relation between pollution price and discharge quantity is as follows:

$$\overline{q}(\theta) = h(\overline{p}, \theta) \tag{22}$$

Suppose $\alpha(\theta), \beta(\eta)$ follows an independent normal distribution.

$$E[\alpha(\theta)] = 0, E[\beta(\eta)] = 0, E[\alpha(\theta)\beta(\eta)] = 0 \tag{23}$$

$$\sigma^2 = E[\alpha(\theta)^2], \sigma_1^2 = E[\alpha(\theta_1)^2] \tag{24}$$

Assumptions $C(q,\theta)B(q,\eta)$ are the actual cost and profit function. Hence, due to the asymmetric information or technical uncertainty, $C(q,\theta_1)$ and $B(q,\eta_1)$ are the costs and benefits of function, the function is estimated by the environmental protection department. $\Delta$ is respect to the gap between estimated values and actual values. $\Delta > 0$ means the environmental protection department's expected revenue is higher than the actual income; $\Delta < 0$ means it is less than the actual income; $\Delta = 0$ means it is equal to the actual income.

$$\Delta = E[(B(\overline{q}(\theta_1),\eta_1) - C(\overline{q}(\theta_1),\theta_1)) - (B(\overline{q}(\theta),\eta) - C(\overline{q}(\theta),\theta))] \tag{25}$$

$$\Delta = \frac{B'' + C''}{2C''^2}(\sigma_1^2 - \sigma^2) \tag{26}$$

Only this condition $\sigma_1^2 = \sigma^2$ or $B'' = -C''$ setup, the estimated function of the environmental protection department is the same as the actual situation. Still, when this condition $\sigma_1^2 \neq \sigma^2, B'' \neq -C''$ does not set up, it is not the same. The environmental protection department either sets emission fees too high, which causes fewer emissions, or sets emission fees too low, which causes many emissions. Although environmental protection departments under the overall revenue and cost function by pricing $\overline{p}$ can get the optimal social pollution emissions $\overline{q}(\theta) = S_T^*$ for $h$ polluters, there is, due to the asymmetric information, no way to estimate a single polluter's cost and benefits accurately. For $h$ polluters, either $\Delta > 0$ or $\Delta < 0$, the decision of the environmental protection department cannot be satisfied with each polluter. Therefore, the emissions fee will lead to the uncertainty of emission pollution. When the uncertainty is small, it will have no impact on pollution control, but pollution control is excellent when it is significant.

Corollary 1: When the source of pollution is significant, the uncertainty of pollution control exists in price policy tools.

### 4.2. Larger Social Cost along with Quantity Policy

The following proof is based on Wang's model. Suppose there are $h$ pollution sources in the economic system. According to the characteristics of the actual cost of emission reduction, the cost function of emission reduction of the $k$th pollution source is:

$$C^k(s) = a_k S^{-\alpha} \tag{27}$$

In the formula, $\alpha$ is the elastic coefficient of control costs related to emissions. Changing $\alpha$ will mean changing the gradient or curve of the cost function. The larger $\alpha$ is, the steeper the curve is. The marginal cost curve of emission reduction rises fast with the decline of emission level. $\alpha$ is the same for all pollution sources, but the coefficient $a_k$ varies with

pollution sources due to pollution sources. The marginal cost curve of emission reduction, the curve of the *k*th pollution source, is obtained.

$$- C^k(s) = MCA_k = -\alpha a_k S^{-\alpha-1} \tag{28}$$

According to the characteristics of the actual pollution damage, we suppose that the linear dose-response model is adopted for pollution damage, then the pollution damage function is:

$$D(S_1, S_2 \cdots S_k) = \lambda \left( \sum_{k=1}^{h} S_K \right) \tag{29}$$

In Equation (30), $\lambda$ is the shadow price of emissions, namely the actual monetary loss caused by the emission unit. According to Equations (27) and (29), the social total cost function model is

$$SC = \sum_{k=1}^{h} a_k s_k^{-\alpha} + \lambda \sum_{k=1}^{h} s_k \tag{30}$$

Taking the derivative of Equation (30) obtains the optimal emission level of each enterprise under the lowest social cost $S_k^*$:

$$S_k^* = \left( \frac{\alpha a_k}{\lambda} \right)^\varepsilon \tag{31}$$

In Equation (31) $\varepsilon = 1/(\alpha+1)$, substituting Equation (31) into Equation (30) obtains the total social cost of all pollution sources at the optimal emission level $SC^*$:

$$SC_{one}^* = \left( \frac{\lambda}{\alpha} \right)^{\alpha\varepsilon} (1+\alpha) \left( \sum_{k=1}^{h} a_k^\varepsilon \right) = \left[ \left( \frac{\lambda}{\alpha} \right)^{\alpha\varepsilon} (1+\alpha) h \right] E(a^\varepsilon) \tag{32}$$

$$\text{Set } E(a^\varepsilon) = \frac{1}{h} \sum_{k=1}^{h} a_k^\varepsilon \tag{33}$$

When the emissions fee $p^*$ equals $\lambda$, the *k*th pollution source minimizes the total expenditure $(a_k S^{-\alpha} + p^* S_k)$. Thus, the optimal emission level $S_k^*$ of each pollution source is Equation (33), when the emissions fee is $p^*$.

$$S_k^* = \left( \frac{\alpha a_k}{p^*} \right)^\varepsilon = \left( \frac{\alpha a_k}{\lambda} \right)^\varepsilon \tag{34}$$

The total emissions are

$$S_T = \sum_{k=1}^{h} S_k^* \tag{35}$$

Now the polluter is required to implement the uniform emissions standard $S_k = \overline{S}(k = 1, 2, \cdots h)$ and the total social cost $SC_{st}$ is

$$SC_{st} = \sum_{k=1}^{h} a_k \overline{S}^{-\alpha} + \lambda \sum_{k=1}^{h} \overline{S} \tag{36}$$

The above formula takes the first derivative $\overline{S}$ to obtain the best uniform emission level $\overline{S}^*$.

$$\overline{S}^* = \left( \frac{\alpha}{\lambda h} \sum_{k=1}^{h} a_k \right)^\varepsilon \frac{1}{h} \sum_{k=1}^{h} a_k \tag{37}$$

Substituting from (37) into (36),

$$SC_{st}^* = \left(\frac{\lambda}{\alpha}\right)^{\alpha\varepsilon}(1+\alpha)\left(\sum_{k=1}^{h} a_k\right)^{\varepsilon} = \left[\left(\frac{\lambda}{\alpha}\right)^{\alpha\varepsilon}(1+\alpha)h\right]E(a)^{\varepsilon}E(a) = \frac{1}{h}\sum_{k=1}^{h} a_k \tag{38}$$

$$\sum_{k=1}^{h} S_k^* = h\overline{S} = S_T \tag{39}$$

$$[E(a)]^{\varepsilon} = \left(\frac{1}{h}\sum_{k=1}^{h} a_k\right)^{\varepsilon} > E(a^{\varepsilon}) = \frac{1}{h}\sum_{k=1}^{h} a_k^{\varepsilon} \tag{40}$$

According to (32), (38), and (40),

$$SC_{one}^* \le SC_{st}^* \tag{41}$$

The total social cost of emission trading is given below. It assumes that the environmental protection department distributes discharge capacity $\overline{S}$ for each polluter according to the amount of ecological balance. If the $k$ polluter discharges more than it, they will be fined. Each polluter can use the next time if they are not used up their quotas within the prescribed period; they can also trade their quotas in the sewage market.

The transaction costs for each polluter are $\Phi(\gamma)$, assuming there are $n$ polluters in the $h$ polluters, whose emissions do not exceed $\overline{S}$, so $h - n$ polluters exceed $\overline{S}$, we get the total social cost, $SC_{ct}$. The total social cost of emissions trading consists of three parts: the cost of each person issuing emission permits and the total cost of pollution supervision, the cost saved by $n$ polluters after emission permit trading, and the transaction cost of emission trading (The second term in the formula is the difference between emissions standard and emissions fee, that is, a certain income can be obtained after selling their emission permit at the price of emissions standard according to their marginal cost of emission.).

$$
\begin{aligned}
SC_{ct} &= \left[\left(\frac{\lambda}{\alpha}\right)^{\alpha\varepsilon}(1+\alpha)h\right]E(a)^{\varepsilon} - \frac{n}{h}\left[\left(\frac{\lambda_1}{\alpha}\right)^{\alpha\varepsilon}(1+\alpha)h\right]\left(E(a)^{\varepsilon} - E(a^{\varepsilon})\right) + h\Phi(\gamma) \\
&= \frac{n}{h}\left[\left(\frac{\lambda}{\alpha}\right)^{\alpha\varepsilon}(1+\alpha)h\right]E(a^{\varepsilon}) + \frac{h-n}{h}\left[\left(\frac{\lambda}{\alpha}\right)^{\alpha\varepsilon}(1+\alpha)h\right]E(a)^{\varepsilon} + h\Phi(\gamma) \\
&= \begin{cases} \left[\left(\frac{\lambda}{\alpha}\right)^{\alpha\varepsilon}(1+\alpha)h\right]E(a)^{\varepsilon} + h\Phi(\gamma), n \to 0 \\ \frac{1}{2}\left[\left(\frac{\lambda}{\alpha}\right)^{\alpha\varepsilon}(1+\alpha)h\right]\left(E(a^{\varepsilon}) + E(a)^{\varepsilon}\right) + h\Phi(\gamma), \ n = \frac{h}{2} \\ \left[\left(\frac{\lambda}{\alpha}\right)^{\alpha\varepsilon}(1+\alpha)h\right]E(a^{\varepsilon}) + h\Phi(\gamma), n \to h \end{cases}
\end{aligned} \tag{42}
$$

If and only if, $\Phi(\lambda)$, $h$ that is relative to the total discharge charge is small, namely,

$$\lim_{\Phi(\gamma)\to 0}\frac{h\Phi(\gamma)}{SC_{ct}} = 0, \lim_{h\to 0}\frac{h\Phi(\gamma)}{SC_{ct}} = 0 \tag{43}$$

The total social cost of emissions trading is between emissions fees and emission standards when $n \to 0$, $n = h/2$, $n \to h$ (See Figure 2).

By (42) and (43),

$$SC_{one}^* \le SC_{ct} \le SC_{st} \tag{44}$$

Corollary 2: When pollution is a minor source, the quantity policy tool will generate enormous total social cost.

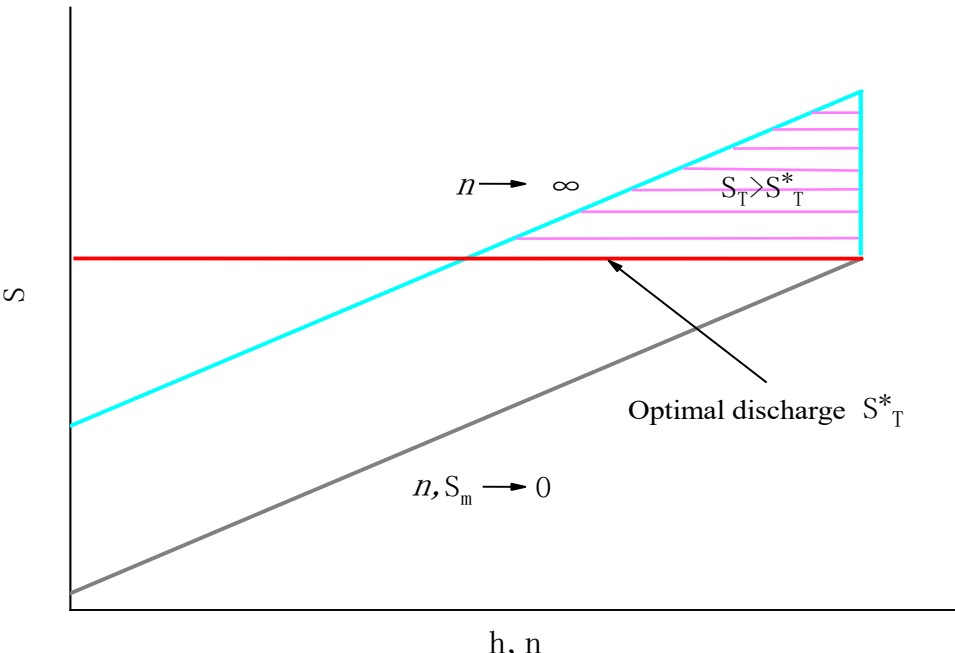

**Figure 2.** The social cost of emissions trading on control of the small source.

### 4.3. The Problem of Emissions Trading

Let's suppose that there are $h$ polluters in the country or the world, which are evenly distributed into two regions, the region $a$ has $h/2$ polluters, and the region $b$ has $h/2$ polluters (The model assumes a uniform distribution, and the conclusions obtained according to its assumptions do not affect the general inference). The optimal emission level planned by the environmental protection department in the whole region is $\overline{S} = S_T^*/h$, of which $S_T^*$ is the ideal emission level in the society, and the emission level of area $a$ and $b$ is

$$S_a^* = S_b^* = h\frac{\overline{S}}{2} = \frac{S_T^*}{2} \tag{45}$$

Assuming that the region $a$ has a lower ecological absorption rate, the environmental protection department of the region $a$ implements stricter emission standards $\overline{S}_H$ for each polluter, while the region $b$ is still $\overline{S}$, and

$$\overline{S}_H < \overline{S}, seting, \overline{S}_H = \frac{S_T^*}{h+2} \tag{46}$$

As the region $a$ carries out stricter standards than region $b$, the emissions of the region $a$ at this time are

$$S_{aH}^* = h\frac{\overline{S}_H}{2} = h\frac{S_T^*}{2(h+2)} \tag{47}$$

Compare the number of emissions before and after the stricter standards:

$$S_a^* - S_{aH}^* = h\frac{\overline{S}}{2} - h\frac{\overline{S}_H}{2} = \frac{S_T^*}{2} - h\frac{S_T^*}{2(h+2)} = \frac{S_T^*}{h+2} \tag{48}$$

Under the strict emissions standard $\overline{S}_H$, the region $a$ has $S_T^*/(h+2)$ emissions gap compared with $S_b^*$ or the previous emission level of the region $S_a^*$. If the permits of the two regions can freely trade, then the total emissions of the two regions are

$$S_a^* + S_b^* = \frac{S_T^*}{2} + \frac{hS_T^*}{2(h+2)} = \frac{S_T^*(h+1)}{h+2} \tag{49}$$

The emissions of the two regions are averaged after emissions trading (Because of the emission gap in the region, the polluter of the region will buy more emission permits than the polluter of the region. The emission permit market will be balanced and stable according to the Walrasian equilibrium. Then the excess emission permits will be evenly distributed in two regions under the market. If the market is not balanced and the gap in the zone is not made up, or if the distribution is not equally divided between the two regions, then the polluter in the zone will continue to buy the license until there is no license. This discharge leakage is based on the same principle of pressure-flow as in physics, where pressure flows from high to low until there is no pressure difference.), and then the emissions of the regions *a* and regions *b* are

$$S_a = S_b = \frac{S_a^* + S_b^*}{2} = \frac{S_T^*(h+1)}{2(h+2)} \tag{50}$$

The comparison of Equations (49) and (50).

$$S_{aH}^* = \frac{2\overline{S}_H}{h} = 2\frac{S_T^*h}{h+2} < S_b = \frac{2S_T^*(h+1)}{h+2} \tag{51}$$

In Figure 3, emissions are leaked from region *b* to region *a* after emissions trading, and the leaked pollution is as follows:

$$\Delta S = \frac{2S_T^*}{(h+2)} \tag{52}$$

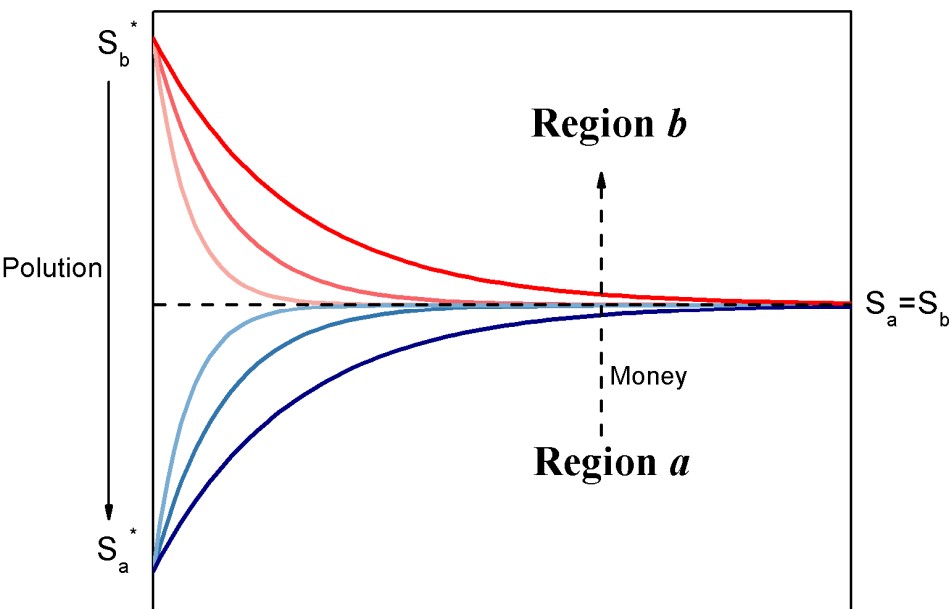

**Figure 3.** Pollution leaks from region *b* to region *a*.

The expenses of leaked pollution are:

$$\Delta SC = \lambda \frac{2S_T^*}{(h+2)} \tag{53}$$

After the pollution leaked, the high emission standard in region *a* will invalidate, and the money will flow the region *a* into the region *b*, namely *ΔSC*, equivalent to an invisible subsidy given from region *a* to region *b*. If the permits of the two regions are not allowed to flow and trade freely, the polluter will transfer from region *a* to region *b*. This

leakage will happen under any environmental control policy, which is also known as the polluter transfer.

Corollary 3: When the speed of ecological absorption is not constant, adjusting the number of part pollution permits will lead to two kinds of pollutant discharge leakage. If the pollution permits can freely circulate, it will lead to the pollution of regional *b* (less regulated areas) inflow into the region *a* (more regulated areas); if it does not freely circulate, the pollution flow has gone into reverse.

## 5. Conclusions, Limitations, and Future Recommendations

### 5.1. Conclusions

This paper proves theoretically that: (1) When the ecological absorption speed is constant, the quantity policy tool (emission standard) is the preferred tool to control significant pollution sources. The price policy tool (emission charge) is the first choice to control minor pollution sources. (2) When the speed of ecological absorption is not constant, adjusting the number of part pollution permits will lead to two kinds of discharge leakage. If the license can freely trade, it will lead to the pollution of the regional *b* (less regulated areas) inflow into the region *a* (more regulated areas); if it does not freely circulate, the pollution flow has gone into reverse. Overall, both policies' wrong application will take high social costs. As minor pollution sources, restaurants, motor vehicles, and agricultural production should use the Pigou tax to control pollution. In contrast, significant pollution sources, such as power plants, cement plants, and paper mills, should use emissions trading to control pollution.

### 5.2. Limitations

There are several limitations to this study. Although many countries use a single policy (Pigou tax or emissions trading) or a mixed policy (Pigou tax and emissions trading) to control pollution, there is a lack of quasi-natural experiments or data to test the above conclusion. Hence, the above findings can hardly be further tested by empirical data in this paper. Second, there are many implicit hypotheses in the theoretical proof, which may apply to specific conditions and generate many conclusions if the conditions change. Due to the discussion of the applicable conditions of the two policies, the differences between the two policies in different countries and economies are not discussed only from the theoretical level, which is a shortcoming of this paper.

### 5.3. Future Recommendations

In the early 20th century, the discussion of Pigouvian taxes and the Coase theorem was not interrupted. Based on the theory of Pigou and Coase, scholars from various countries proposed emissions trading and emission fees, respectively, which solved the complicated environmental problems in many countries. Regardless of whether any economic theory has its advantages and disadvantages, the ecological and economic system difference bring difficulties to applying economic theory. Countries should use related policies to control environmental pollution under other circumstances to make the policies consistent, feasible, and scientific in theory and practice. In future research, we will further improve the defect of Pigou's tax, replacing one-part charging of emissions with two-part charging of emissions to make up for the uncertainty in total pollution control. To further illustrate the role of two-part charging of emissions, we will compare the part of the two-part charging of emissions and emissions trading in pollution control.

**Author Contributions:** Conceptualization, Q.W. and X.Z.; methodology, X.Z.; software, Q.W. and J.H.; validation, J.H., H.Z. and Y.D.; formal analysis, J.H.; investigation, H.Z.; resources, X.Z.; data curation, W.Q.; writing—original draft preparation, W.Q; writing—review and editing, Q.W.; visualization, X.Z.; supervision, Y.D.; project administration, J.H.; funding acquisition, Q.W. All authors have read and agreed to the published version of the manuscript.

**Funding:** This research is funded by the Center for Energy and Environmental Policy Research, Chengdu University of Technology.

**Institutional Review Board Statement:** Not applicable.

**Informed Consent Statement:** Not applicable.

**Data Availability Statement:** Data sharing not applicable.

**Conflicts of Interest:** The authors declared no potential conflicts of interest with respect to the research, authorship, and publication of this article.

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
