# Peer review of "Climate Change and Human Response to Sustainable Environmental Governance Policy: Tax or Emissions Trading?"

_sustainability, doi:10.3390/su14159412_

Round 1

Reviewer 1 Report

Dear Authors,

More remarks you can find in attached file. Please take them into consideration.

Author Response

Thank you for your valuable suggestions on the article. We have modified and improved the paper according to your requests. Our reply and modification are as follows:

1.Model

(1) We did not clearly explain the flow pollution and stock pollution in the assumptions derived from the model, and there was confusion in the editing formula. We have made adjustments; please refer to the revised original text for specific adjustments. In the paper we adopt the measurement method of total pollution, that the total amount is divided into flow pollution and stock pollution according to dissipation of pollution rather than discharge representing flow pollution and concentration is instead of stock pollution. is the stock pollution, andis the flow pollution in the model.

(2) Eqs.(1) is the increase of flow pollution, that is, the increase of flow pollution;

(Emssions)                             (1)

Eqs. (2) is the absorption of flow pollution, that is, the reduction of flow pollution

(absorption )                          (2)

We are combining Eqs. (1) and Eqs. (2), we obtain the formula of changes in flow pollution, which mainly construct the Hamilton function. In the Eqs.(1) and Eqs.(2) the meaning of  is using an amount of energy.

(3) is the stock pollution at time 0,is the stock pollution of time,which represents the value of the stock at a point in time. So let's assume it's a constant.

(4) We have reunified the terminology as you suggesttion about the 'absorption'.

2 Price VS. Quantity on controlling signiant (minor) pollution sources

(1) The problem you raised is also one that we have been trying to best solve in the process of deriving the model, but we have not solved it yet after a long discussion, which is mainly two reasons:

  • Firstly, in economic models, it is customary to replace things with generic names such as pollution rather than specific pollutants such as sulfur dioxide and carbon dioxide. So we also describe this treatment as significant and minor polluters rather than establish a criterion to distinguish one class from another.
  • Secondly, the establishment of theoretical model can be further promoted in practice. Only when the condition of theoretical model is established can it be further popularized and applied in practice. Just like the creation of environmental tax and the design of emission trading, the theory was gradually born and gradually applied to practice. The paper mainly discusses the applicable conditions of large and small pollution from the theoretical level, and we indirectly ignore the specific use range of large and small pollution sources. The paper hopes to discuss these problems in detail in the following article.

(2) We have modified wrong expression (Price and Quantity in Figure 1 and two expressions 《Jinnan Wang's Model》 and 《unit Emission》)

  1. Conclusion, Limitations, and Future Recommendations

The paper's primary purpose is to theoretically discuss the applicable conditions of the two policies in different situations to provide theoretical support for applying this policy in practice. Therefore, there may be many problems in practical application,we hope to research real conditions to verify the theory in the following article.

Reviewer 2 Report

The paper " Climate change and human response of sustainable environmental governance policy: Tax or Emissions trading? " is to study Pigou tax vs. Emissions trading from a dynamic perspective and propose an application range to control significant and minor pollution sources. The paper is well written and structured. It should be published after proper revision:

(1) The author mentioned the Pigou tax may suitable for pollution control from minor pollution sources, restaurants, motor vehicles, and agricultural production, while the emission trading is better for pollution control from such as power plants, cement plants, and paper mills. The author may discuss more on the choice of the instrument in different countries, such as China, USA, and India, which are top three emitters in the world. The choice of the instrument on emission trading or carbon tax may depend on national and economic circumstances as well.

(2) It would be beneficial to have the paper proofread to improve the English.

Author Response

  1. Due to the discussion of the applicable conditions of the two policies, the differences between the two policies in different countries and economies are not discussed only from the theoretical level, which is a shortcoming of this paper. Therefore, this deficiency is put into the outlook, hoping to test the application conditions of this theory in different countries through experience.
  2. We have carefully revised the English grammar further, and the specific modifications mark in the markup version.

Round 2

Reviewer 1 Report

Dear Authors,

However I am not economist, I think that your results presented in the paper are convincing enough to publish it in Sustainability.